# Effect of Nanocrystallization of Anthocyanins Extracted from Two Types of Red-Fleshed Apple Varieties on Its Stability and Antioxidant Activity

**DOI:** 10.3390/molecules24183366

**Published:** 2019-09-16

**Authors:** Jihua Xu, Xinxin Li, Shifeng Liu, Peilei Zhao, Heqiang Huo, Yugang Zhang

**Affiliations:** 1Qingdao Key Laboratory of Genetic Development and Breeding in Horticultural Plants, Qingdao Agricultural University, Qingdao 266109, China; xujihua@qau.edu.cn (J.X.); lixinxinll@163.com (X.L.); liu18306390438@163.com (S.L.); zhaopeileiyx@163.com (P.Z.); 2College of Life Sciences, Qingdao Agricultural University, Qingdao 266109, China; 3College of Horticulture, Qingdao Agricultural University, Qingdao 266109, China; 4Mid-Florida Research and Education Center, University of Florida, Apopka, FL 32703, USA; hhuo@ufl.edu

**Keywords:** red-flesh apple, anthocyanins, corn starch nanoparticles, antioxidant activity

## Abstract

Red-fleshed apple (*Malus sieversii f. neidzwetzkyana* (Dieck) Langenf) has attracted more and more attention due to its enriched anthocyanins and high antioxidant activity. In this study we extracted total anthocyanins and phenols from two types of red-fleshed apples—Xinjing No.4 (XJ4) and Red Laiyang (RL)—to study the stability and antioxidant activity of anthocyanins after encapsulation onto Corn Starch Nanoparticles (CSNPs). The results indicated the anthocyanins and total phenol levels of XJ4 were 2.96 and 2.25 times higher than those of RL respectively. The anthocyanin concentration and loading time had a significant effect on CSNPs encapsulation, and XJ4 anthocyanins always showed significantly higher loading capacity than RL. After encapsulation, the morphology of RL-CSNPs and XJ4-CSNPs was still spherical with a smooth surface as CSNPs, but the particle size increased compared to CSNPs especially for RL-CSNPs. Different stress treatments including UV light, pH, temperature, and salinity suggested that XJ4-CSNPs exhibited consistently higher stability than RL-CSNPs. A significantly enhanced free radical scavenging rate under stress conditions was observed, and XJ4-CSNPs had stronger antioxidant activity than RL-CSNPs. Furthermore, XJ4-CSNPs exhibited a slower released rate than RL-CSNPs in simulated gastric (pH 2.0) and intestinal (pH 7.0) environments. Our research suggests that nanocrystallization of anthocyanins is an effective method to keep the anthocyanin ingredients intact and active while maintaining a slow release rate. Compared to RL, encapsulation of XJ4 anthocyanins has more advantages, which might be caused by the significant differences in the metabolites of XJ4. These findings give an insight into understanding the role of nanocrystallization using CSNPs in enhancing the antioxidant ability of anthocyanins from different types of red-fleshed apples, and provide theoretical foundations for red-fleshed apple anthocyanin application.

## 1. Introduction

In recent years, many red-fleshed apple (*Malus sieversii f.neidzwetzkyana* (Dieck) Langenf) cultivars have been reported and have attracted attention from researchers and breeders for both visual and nutritional reasons [1,2,3]. Red-fleshed apple is enriched in anthocyanins and recognized as a functional fruit [4,5]. The observed red coloration in red-fleshed apple is the consequence of anthocyanins accumulation. Anthocyanins are a class of flavonoid compounds, mainly including pelargonidin, cyanidin, delphinidin, peonidin, malvidin, and petunidin [6]. Functioning as antioxidants, anthocyanins play an important role in providing nutritional value, maintaining normal vascular permeability and protecting human against diseases such as hyperglycemia and cancer [7,8,9,10].

The chemistry characteristics of anthocyanins have been studied extensively. Anthocyanins are water-soluble pigments with many phenolic hydroxyl groups that can interact with proteins to inhibit peroxidation [11,12,13]. In addition, earlier studies have reported anthocyanins pigments are involved in reducing lipid peroxidation and scavenging free radicals [14]. However, it is difficult for anthocyanins to remain active and functional during food processing or storage, as anthocyanins are sensitive to environmental factors. Under the influence of temperature, pH, UV light, and metal ions, anthocyanins become unstable and show degradation due to the chemical constitution changes [15,16].

The encapsulation of anthocyanins could be an effective method to hinder bioactive compounds from oxidation and degradation in sensitive environments. In our previous study, we explored the use of biocompatible zein to encapsulate anthocyanins and found the nanoencapsulation could enhance its stability and improve its antioxidant effect [17]. Compared to zein, Starch Nanoparticles (SNPs) are more natural and common but using starch as carriers to load and deliver anthocyanins from red-fleshed apple have not been reported. SNPs use natural starch as carriers to load and deliver bioactive compounds. The properties of biocompatibility—a size less than 300 nm, and a large surface—make SNPs promising nanocarriers to be broadly used in food, drugs, and cosmetics [18,19]. SNPs can encapsulate bioactive compounds such as polyphenols by hydrogen bonds and hydrophobic interactions, improving aqueous solubility. The encapsulation of polyphenolic compounds can hinder them from oxidation and degradation in sensitive environments. Nanoparticles with a small size can be directly taken by epithelial cells in the small intestine, further enhancing the absorption and bioavailability of bioactive compounds [20,21]. The application of SNPs to encapsulate essential oils was studied previously [22]. It was found that when the complexation temperature was 90 °C, the encapsulation efficiency achieved a maximum of 86.6%. The encapsulation of essential oils in SNPs maintains a slow release rate of essential oils and significantly enhances stability and antioxidant activity. Another research was related the application of SNPs to incorporate four types of polyphenols. The adsorption kinetics, loading amount and efficiency, in vitro release and antioxidant activity were investigated. The results indicated SNPs-carried polyphenols showed a sustained release character and stability under adverse environmental conditions [23].

The materials we used in this study were two red-fleshed apple varieties, RL and XJ4, with abundant anthocyanin content, selected from our breeding work [2,4]. The promoter region of *MdMYB10* is different in different types of red-fleshed apples, and the R6 promoter element is strongly related with red-fleshed fruit. The genotype of red-fleshed apple can be a heterozygous R1R6 type or a homozygous R6R6 type while white-fleshed apple is an R1R1 type [24], and in our previous work we detected that RL is an R1R6 type and XJ4 is an R6R6 type (data unpublished). Our previous study compared the encapsulation capacity of SNPs prepared from corn, sweet potato, and potato and found that Corn Starch Nanoparticles (CSNPs) had the highest encapsulation efficiency (data unpublished). Although using CSNPs as nanocarriers for encapsulation has been widely reported, less was known with regard to using CSNPs as a carrier to encapsulate anthocyanins extracted from red-fleshed apples. Therefore, in this study we used CSNPs to nanocrystallize anthocyanins obtained from two red-fleshed apple varieties, aiming to investigate the effect of anthocyanin nanoencapsulation on its stability, antioxidant activity, and release properties in the two varieties. Our results will provide a theoretical basis and technical support for how to protect anthocyanins of red-fleshed apples from degradation.

## 2. Results and Discussion

### 2.1. The Anthocyanins and Total Phenols Contents of the Two Types of Red-fleshed Apple

The phenotypes of the two types of red-fleshed apple are different (Figure 1A). The skin color of RL is light red, and the outer flesh is red with the core remaining white, while XJ4 has purplish red skin and bright red flesh. We further detected the anthocyanins and total phenol contents in the two types of red-fleshed apple (Figure 1B). The anthocyanins and total phenol levels in the R1R6 type RL were 141.1 mg/kg (FW) and 1015.3 mg/kg (FW) respectively, while in the R6R6 type XJ4 the anthocyanins and total phenols levels were 2.96 and 2.25 times higher than that of RL, with 418.6 and 2286.3 mg/kg (FW), respectively. The anthocyanin contents were found to be consistent with the differences in the skin and flesh colors between these two varieties. In addition, through the comparison of metabolomics data of RL and XJ4, we obtained 46 types of statistically significant different metabolites, belonging to seven groups including anthocyanins, flavones, flavonoids, polyphenols, flavanones, isoflavones, and flavonols (Appendix A). Among the 46 types of metabolites there were nine that showed very significant differences (*p* < 0.001), such as Cyanidin 3-*O*-malonyl hexoside, Syringetin 5-*O*-hexoside, Syringetin 7-*O*-hexoside, and so on. Therefore, the detected and very significantly different metabolites in these two varieties provides a theoretical foundation for the comparison of anthocyanin stability and antioxidant activity between the RL and XJ4 anthocyanin corn starch hybrid nanoparticles in the subsequent study.

### 2.2. The Comparison of Encapsulation Properties of the Two Types of Red-fleshed Apple using CSNPs

#### 2.2.1. The Effect of Different Anthocyanin Concentrations on CSNPs Encapsulation

CSNPs were prepared by nanoprecipitation to encapsulate the anthocyanins in the two types of red-fleshed apple, and the effect of different anthocyanin concentrations on CSNPs encapsulation was explored. We tested a series of concentrations from 20 to 100 mg/L (Figure 2). The maximum concentration, 100 mg/L, was chosen based on the anthocyanins content of RL (only 141.1 mg/kg). The increase of loading capacity was observed with the increasing concentration of anthocyanins. In both XJ4 and RL, the maximum loading capacity was found when the concentration of anthocyanins was the highest. The XJ4 anthocyanin loading capacity was 81.73 ug/mg, which was 1.77 times higher than RL (46.23 ug/mg). Therefore, the concentration 100 mg/L was chosen for further research. In addition, the XJ4 anthocyanins always showed a significantly higher loading capacity than RL under the same concentration levels.

#### 2.2.2. The Effect of Different Loading Times on CSNPs Encapsulation

The effect of loading time on CSNPs encapsulation was studied when the anthocyanin concentration was 100 mg/L. The results indicated the CSNPs loading capacity was raised with an increase of loading time, until an equilibrium was achieved when the loading time was around 360 min (Figure 3). A dramatic increase in loading capacity was observed during the initial 120 min, and after that time the loading rate was slower. Our observation of the quick increase of loading capacity at the first stage, then the slowed-down loading rate, and finally a plateau, is consistent with previous studies [23,25]. The first stage quick increase is due to the existence of plenty of reactive sites on the CSNPs surface for combining with anthocyanins. The equilibrium at the last stage may be due to the saturation of these binding sites. In addition, the XJ4 anthocyanins always exhibited significantly higher CSNPs loading capacity than RL at the same loading time, around 1.3–1.8 times higher than RL. Finally after 24 h, the CSNPs loading capacity for XJ4 anthocyanins was 85.22 ug/mg. By contrast, the CSNPs loading capacity for RL anthocyanins was 53.14 ug/mg. The higher CNSPs loading capacity of XJ4 anthocyanins might have been caused by the extremely significant high content of Cyanidin 3-*O*-malonyl hexoside in XJ4 (Appendix A), which might provide more binding sites for CNSPs through hydrogen bonds and hydrophobic interactions.

### 2.3. The Effect of Different Anthocyanins Concentrations and Encapsulating Times on Anthocyanins Corn Starch Hybrid Nanoparticles

With the increase of anthocyanins, the anthocyanins corn starch hybrid nanoparticles size was gradually increased, especially for RL anthocyanins Corn Starch Hybrid Nanoparticles (RL-CSNPs), which showed a significant increase (from 115 to 170 nm), while the XJ4 anthocyanins Corn Starch Hybrid Nanoparticles (XJ4-CSNPs) size did not increase significantly (from 95 to 112 nm). The maximum sizes of both RL-CSNPs and XJ4-CSNPs were observed at anthocyanin concentration 100 mg/L, and the RL-CSNPs size tended to be larger than the XJ4-CSNPs size at the same anthocyanin levels (Figure 4A). Therefore, the concentration of 100 mg/L was selected for further loading time study. We found both RL-CSNPs and XJ4-CSNPs sizes had the tendency of a quick increase first, followed by a slow decrease (Figure 4B). The maximum sizes of both RL-CSNPs and XJ4-CSNPs were detected when the loading time was 60 min, reaching up to 161 and 119 nm, respectively. After 60 min, the size of both RL-CSNPs and XJ4-CSNPs decreased slowly, finally declining to 90 and 98 nm, respectively. The increase of size might be because, at the beginning, the nanoparticles aggregated together, and the decrease might be caused by nanoparticle dispersion under the magnetic stirring. In addition, the RL-CSNPs size was larger than the XJ4-CSNPs size at the same loading time.

### 2.4. Transmission Electron Microscopy Images of Anthocyanin Corn Starch Hybrid Nanoparticles

We selected RL-CSNPs and XJ4-CSNPs with a anthocyanin concentration of 100 mg/L and an encapsulating time at 60 min for a Transmission Electron Microscopy (TEM) investigation (Figure 5). The TEM results demonstrated that when uncoated with anthocyanins, the CSNPs were spherical with smooth surface, and a diameter ranging from 60 to 70 nm with molecules was aggregated (Figure 5A). By contrast, after encapsulation with anthocyanins, the shape of the nanoparticles remained spherical with smooth surface, but the nanoparticle size increased and the distribution was homogeneous (Figure 5B,C). The diameter of the RL-CSNPs reached from approximately 115 to 170 nm (Figure 5B), and the diameter of the XJ4-CSNPs from 90 to 110 nm (Figure 5C), which were both larger than the non-encapsulated CSNP on average.

### 2.5. The Stability of RL-CSNPs and XJ4-CSNPs under Different Stress Treatments

Stress factors such as UV light irradiation, low/high pH, high temperature, and high salinity have an effect on the stability of nanoparticles, and further affect the particle size and polymerization dispersion index (PDI). Here we determined the changes of particle size and PDI under these four different types of sensitive environments.

#### 2.5.1. The Effect of UV Light

With the increase of UV irradiation, the particle size and PDI of both RL-CSNPs and XJ4-CSNPs showed the tendency to increase (Figure 6A). When the exposure time was between 2–6 h, the particle size and PDI of both RL-CSNPs and XJ4-CSNPs raised dramatically. Especially, the RL-CSNPs size increased from 174.2 to 322.3 nm, which was greater than the XJ4-CSNPs (from 97.1 to 178.6 nm). In addition, under the same UV light exposure time, the diameter of RL-CSNPs was always larger than the diameter of the XJ4-CSNPs.

#### 2.5.2. The Effect of pH

Different pH levels of the solution provided various acid–alkali environments, which affected the stability of anthocyanins. At pH 1, the maximum particle size was observed, which were 346.1 and 252.4 nm for RL-CSNPs and XJ4-CSNPs, respectively (Figure 6B). In an acidic environment (pH 1~7), with an increase of pH the particle size and PDI of both RL-CSNPs and XJ4-CSNPs decreased markedly. This suggests that anthocyanins tend to be soluble in acidic conditions, which makes it easier to be encapsulated by CSNPs. Thus, the diameter of nanoparticles was large. By contrast, in alkali conditions (pH 9~13), slow changes in particle size and PDI were found. The PDI of both RL-CSNPs and XJ4-CSNPs were relatively high in acidic (pH 1~3) and alkali (pH 9~13) conditions, while when pH was 5~7 the PDI were low. The results indicated that RL-CSNPs and XJ4-CSNPs showed homogenous dispersion under weak acid and neutral conditions.

#### 2.5.3. The Effect of Temperature

The effect of temperature on nanoparticles size was similar to the results of UV light effects. With an increase of temperature (20~100 °C), the particle sizes of both RL-CSNPs and XJ4-CSNPs showed the tendency to increase (Figure 6C). The observation was similar to the study in proanthocyanide in SNPs [26]. The RL-CSNPs size increased from 125.1 to 325.4 nm, which was greater than the XJ4-CSNPs. The XJ4-CSNPs size increased by 76.7 nm when the temperature increased from 80 to 100 °C, while there was no significant change when temperature was below 80 °C. The effect of temperature on PDI in both RL-CSNPs and XJ4-CSNPs was similar to the results of the pH effect, which showed a sudden decrease followed by a slow increase. When at 60 °C, the lowest PDIs were observed in both RL-CSNPs and XJ4-CSNPs—0.21 and 0.24, respectively. This suggests that 60 °C is the most appropriate temperature for RL-CSNPs and XJ4-CSNPs dispersion.

#### 2.5.4. The Effect of Salinity

When the NaCl concentration was less than 0.2 mM, its effect on RL-CSNPs and XJ4-CSNPs was not significant. When the concentration was more than 0.4 mM, we observed the increase of nanoparticle size with the increase of NaCl concentration. Especially, the RL-CSNPs diameter rose from 145.6 to 455.1 nm, which was greater than the rise in XJ4-CSNPs diameter (116.7–299.2 nm). The increase tendency of PDIs was also detected in RL-CSNPs and XJ4-CSNPs under NaCl treatment, with the maximum level (0.58 nm and 0.47 nm, respectively) observed at the highest NaCl concentration. The tendency of size increase with the increase of NaCl content was similar to the finding in a polyphenol SNPs study [23].

After the conducting and analyzing of different stress treatments including UV light, pH, temperature, and salinity, we found their effects on particle size and PDI of RL-CSNPs and XJ4-CSNPs. Combining all the results here, we conclude that XJ4-CSNPs show consistently higher stability than RL-CSNPs under all four different stress treatments. The higher stability of XJ4-CSNPs under different stress treatments makes it a potential antioxidant. Therefore, in the following research, we studied the antioxidant capacity of XJ4-CSNPs when compared to RL-CSNPs.

### 2.6. The Antioxidant Activity of RL-CSNPs and XJ4-CSNPs under Different Stress Treatments

The anthocyanins are very unstable and easily degradable under UV light, high/low pH, high temperature and high salinity, which further affects its antioxidant activity. However, after encapsulating the anthocyanins in CSNPs to form the nanocomplexes, the radical scavenging activity was significantly enhanced (Figure 7). Under UV light, the DPPH free radical scavenging rate was markedly enhanced after encapsulation (RL-CSNPs and XJ4-CSNPs, Figure 7A). As the exposure time increased, the RL and XJ4 anthocyanin extract solution scavenging rates showed sharp decreases, dropped down from 81.3% and 89.1% to 28.9% and 56.6%, respectively. As the RL and XJ4 anthocyanins were encapsulated to form RL-CSNPs and XJ4-CSNPs nanocomplexes, the scavenging rates showed a slow increase and then a slow decrease. The increase of scavenging rates in the first two hours was probably due to nanoparticles capturing more energy under UV irradiation, and releasing more anthocyanins, thus the release rate of anthocyanins was higher than the degradation rate. After two hours, the scavenging rate started to decrease—the scavenging rate of RL-CSNPs and XJ4-CSNPs declined to 76.4% and 79.1%, respectively, after six hours of UV irradiation—which showed significant higher antioxidant activity compared to the non-encapsulated anthocyanins of RL and XJ4.

Both strong acid and alkali treatments reduced RL and XJ4 anthocyanin antioxidant activity, especially when pH was from 9 to 13 (Figure 7B). After encapsulation, RL-CSNPs and XJ4-CSNPs exhibited significantly enhanced abilities of DPPH free radical scavenging rates in strong acid and alkali conditions; especially in the strong alkali treatment, the radical scavenging activity was 1.4–2.7 fold higher than the non-encapsulated anthocyanin extract solutions. Similar observation was reported in our previous study, which found that after encapsulating anthocyanins from red-fleshed apple with zein, the antioxidant activity was greatly enhanced under alkaline treatments (pH 9.0) [17].

The effects of temperature on RL and XJ4 anthocyanin extract solutions, RL-CSNPs and XJ4-CSNPs, were different (Figure 7C). A significant scavenging rate difference was observed in anthocyanins extract solutions under higher temperatures. When the temperature was raised from 20 to 100 °C, the scavenging rates of RL and XJ4 dropped to 63.4% and 70.2% from 82.1% and 87.3%, respectively. There was no obvious change for the scavenging rates of RL-CSNPs and XJ4-CSNPs from 20 to 60 °C until the temperature was higher than 60 °C whereby the scavenging rates started to decrease. Finally, when the temperature reached 100 °C, the scavenging rates of RL-CSNPs and XJ4-CSNPs declined to 77.3% and 84.5%, respectively, which were higher than observed in RL and XJ4 (63.4% and 70.2%). The enhanced scavenging rates especially under higher temperatures after encapsulation of anthocyanins were also reported in our previous study [17].

In higher salinity solutions, RL and XJ4 anthocyanin extract solutions, RL-CSNPs and XJ4-CSNPs, showed lower levels of scavenging rates (Figure 7D). The scavenging rates of RL and XJ4 declined by 23.8% and17.3%, while the scavenging rates of RL-CSNPs and XJ4-CSNPs dropped by 9.1% and 9.8%. In conclusion, after the encapsulation of RL and XJ4 anthocyanins, the RL-CSNPs and XJ4-CSNPs showed enhanced antioxidant activity under all four different types of stress treatments. This might because the nanocrystallization improve anthocyanin stability and provides an additional layer of protection from anthocyanin degradation [27]. The nanocomplex XJ4-CSNPs is a stronger antioxidant compared to RL-CSNPs. This is probably due to the differences in promoter regions of *MdMYB10* caused by the varied anthocyanin levels in different types (R1R6 and R6R6) of red-fleshed apples. The increased expression of *MdMYB10* promoted by the R6 element in XJ4 resulted in more downstream anthocyanins and flavones synthesis; for example, the anthocyanins member cyanidin 3-*O*-malonyl hexoside and flavones member syringetin 5-*O*-hexoside, syringetin 7-*O*-hexoside, butin and so on were only detected in XJ4 (Appendix A). The higher level of downstream metabolites in XJ4, especially the anthocyanins group, might have contributed to the stronger antioxidant activity of XJ4-CSNPs.

### 2.7. The in Vitro Release Rate of Anthocynins after Nanocrystallization

The release experiment was simulated in vitro using dialysis bags. In the first two hours of the whole in vitro release process, the non-encapsulated anthocyanins RL and XJ4 release rate increased markedly (from 51% to 72%) in the simulated gastric (pH 2.0) and intestinal (pH 7.0) environment; while after encapsulation, the RL-CSNPs and XJ4-CSNPs were released slowly (from 31% to 37%) (Figure 8). The non-encapsulated anthocyanins release rate reached a plateau (81%) after four hours in gastric conditions (Figure 8A) and after five hours in an intestinal condition (Figure 8B). At the same release time point, the encapsulated RL-CSNPs and XJ4-CSNPs release rate was only 50–53%, as the nanoparticles continued to diffuse the release rate by slow increases. The release rate of RL-CSNPs and XJ4-CSNPs reached the level of RL and XJ4 when the release time was 12 h. The release rate of encapsulated anthocyanins was faster at pH 2.0 than at pH 7.0, indicating that, in an acidic environment, the binding affinity of anthocyanins with CSNPs was weaker than in an alkali environment. This result is consistent with a previous study in protein–proanthocyanidins hybrid nanoparticles [26]. Compared to RL-CSNPs, XJ4-CSNPs were released slower at both pH 2.0 and pH 7.0 conditions, suggesting that the intermolecular interaction and surface forces of XJ4 anthocyanins and CSNPs were stronger than RL and CSNPs. This in vitro release study revealed that nanocrystallization of anthocynins had the expected effect of reducing the release rate.

## 3. Material and Methods

### 3.1. Materials

Two types of red-fleshed apples “Red Laiyang” (RL) and “Xinjiang No.4” (XJ4) were grafted on *Malus Robusta* for 6 years at the experimental farm of Qingdao Agricultural University (Qingdao, China). The ripen fruits (120 days after flowering) were collected and put into ice box before transferring them to a laboratory. Corn starch, sweet potato starch, and potato starch were purchased from Tianjing Dingfeng Starch Co., Ltd. (Tianjing, China).

### 3.2. Measurement of Total Anthocyanins and Total Phenols Content

The anthocyanins and phenols extraction from the peel and flesh of two types of red-fleshed apples was conducted with absolute ethanol at the ratio of 1:10 (fruit tissues to extraction buffer), kept under dark and 4 °C conditions for 10 h. The supernatant was subsequently filtered with a 0.45 µm membrane and stored at −4 °C. Total anthocyanin content was detected following the pH differential spectroscopic method [28]. The procedure of total anthocyanin determination was exactly the same as in our previous publication by Zhang [17]. The total phenol content was detected following the Folin–Ciocalteu method, with modifications by Cai [29]; the procedure was the same as described by Zhang [17].

### 3.3. Preparation of CSNPs

The CSNPs were prepared by the method of nanoprecipitation [18]. One gram of corn starch was dissolved in 100 mL deionized water, heated and stirred for 30 min in boiling water until the starch was fully gelatinized. The amount of 400 mL absolute ethanol was added to the cooling down gelatinized starch solution, and stirred for 10 h by a magnetic stirring apparatus. After 10 min centrifuging at 3500 rpm, absolute ethanol was used to wash the suspensions three times, and then freeze-dried for 48 h to obtain CSNPs.

### 3.4. Preparation of RL-CSNPs and XJ4-CSNPs

The RL-CSNPs and XJ4-CSNPs preparation was conducted according to Phan’s method [25]. The amount of 10 mg CSNPs was dissolved in 10 mL RL and XJ4 anthocyanin extract solution with a series of concentrations (20, 40, 60, 80, and 100 mg/L). The mixture was shaken for 12 h at 18 °C to make CSNPs fully incorporate into anthocyanin extract. After encapsulation, the nanoparticles suspensions were centrifuged for 10 min at 3500 rpm. The sediment was washed twice and freeze-dried to obtain RL-CSNPs and XJ4-CSNPs, and then stored at 4 °C for further analysis. The supernatant was collected to calculate the encapsulating capacity. Encapsulating capacity (ug/mg) = (total anthocyanins content − anthocyanins content in supernatant)/total weight of dry RL-CSNPs or XJ4-CSNPs. When the anthocyanin concentration was 100 mg/L, the effect of anthocyanin encapsulating time on encapsulating capacity and particles size was investigated, the timing points were selected at 10, 30, 60, 120, 360, 480, 720, and 1440 min.

### 3.5. Measurement of Particles Size and PDI

Napoparticles size and PDI were measured by a Zetasizer Nano ZS90, Malvern Instruments (Malvern, UK). The total amount of a 1 mL sample solution was put into the test room and detected by the dynamic lighting scattering technique.

### 3.6. Transmission Electron Microscopy (TEM)

The morphological images of CSNPs, RL-CSNPs and XJ4-CSNPs, were investigated by a Hitachi 7700 transmission electron microscope (TEM) (Tokyo, Japan), with the accelerated voltage at 80 KV. The samples were suspended in deionized water with 15 min sonication. Afterwards, a droplet of the suspension was added on copper grid coated film and freeze-dried for observation and measurement.

### 3.7. Stress Treatments of RL-CSNPs and XJ4-CSNPs

The intensity of UV light irradiation was 70 uW/cm^2^, and exposure time was 0 h, 0.5 h, 1 h, 2 h, 4 h, and 6 h. In the acidic and alkali treatments, pH was set as 1, 3, 5, 7, 9, 11, and 13 and samples were kept in the dark for 30 min. The temperature treatment was conducted by keeping samples in a water bath of 20, 40, 60, 80, and 100 °C for 30 min. The salinity treatment was conducted with 0, 0.2, 0.4, 0.6, 0.8, and 1 mM NaCl for 30 min.

### 3.8. Measurement of Scavenging Capacity

The antioxidant activity experiment was conducted through determining the scavenging capacity of DPPH free radicals. The method of Yi [30] was used with modification. The RL and XJ4 anthocyanin extracts (100 mg/L) and CSNPs suspension (1000 mg/L) were used as control. The amount of a 1 mL sample was fully mixed with 2 mL of 0.2 mmol/L DPPH, and the mixture was left at room temperature for 30 min. The absorbance was detected at 515 nm with a spectrophotometer. The scavenging rate was calculated as: Scavenging rate (%) = [(DPPH_t0_ − DPPH_t1_)/DPPH_t0_] × 100%. DPPH_t0_ means the absorbance at the beginning and DPPH_t1_ means the absorbance after 30 min of reaction.

### 3.9. Measurement of in Vitro Release Rate of Anthocyanins

The in vitro release rate of anthocyanins after nanocrystallization was examined by the equilibrium analysis method [18]. The simulated gastric and intestinal environment was presented by phosphate buffer saline with pH 2.0 and pH 7.0, respectively. The RL-CSNPs and XJ4-CSNPs were dissolved in phosphate buffer saline and later added into a dialysis membrane with sealed ends and molecular weight cut-off 5 KDa. The dialysis bag was fully immerged into phosphate buffer saline and dialyzed with magnetic stirring at room temperature. The amount of 1 mL released anthocyanins solution was taken out and added in the same amount of fresh buffer at various time points (0–12 h).

### 3.10. Sample Preparation, Metabolite Identification and Quantification

The freeze-dried peel and flesh of RL and XJ4 were crushed by a mixer mill (MM400, Retsch, Haan, Germany) using zirconia beads at 30 Hz for 1.5 min. The powder (100 mg) was weighted and mixed with 1.0 mL 70% aqueous methanol for overnight extraction at 4 °C. Then the extracts were centrifuged at 10,000× *g* for 10 min. Prior to liquid chromatograph/mass spectrometer (LC/MS) analysis, the extracts of peel and flesh were absorbed (CNWBOND Carbon-GCB SPE Cartridge, Shanghai, China) and filtered (SCAA-104, Shanghai, China). The amount of a 5 μL sample was used in a LC–ESI–MS/MS system (HPLC, Shim-pack UFLC SHIMADZU CBM30A system, MS, Applied Biosystems 6500 Q TRAP) for analysis. The analytical conditions were set as described by Wang [31]. Metabolite identification was carried out according to MWDB (metware database), MassBank (http://www.massbank.jp/), KNAPSAcK (http://kanaya.naist.jp/KNApSAcK/), HMDB (http://www.hmdb.ca/), METLIN (http://metlin.scripps.edu/index.php), MoTo DB (http://www.ab.wur.nl/moto/). Metabolite quantification was conducted using MRM. The variable importance in projection (VIP) > 1 and fold change > 2 (upregulation) or < fold change 0.5 (down regulation) was set to filter metabolites with significant differences.

### 3.11. Statistical Analysis

All the experiments were conducted with three replicates. The mean and standard deviations of experimental data were calculated. Analysis of variance (ANOVA) test or Student’s *t* test were conducted for significant differences (significance level of 95%, *p* < 0.05). Multiple comparisons and plot charts were made using Graph-Pad prism 5 analysis software and Origin 9.0 software (OriginLab, Northhampton, MA, USA).

## 4. Conclusions

The results indicated different anthocyanin and total phenol contents in two types of red-fleshed apples, RL and XJ4, with XJ4 showing significantly higher levels. XJ4 anthocyanins also exhibited significantly higher loading capacity. XJ4-CSNPs showed consistently higher stability and enhanced antioxidant activity than RL-CSNPs under different stress treatments. Furthermore, XJ4-CSNPs had a slower released rate than RL-CSNPs in simulated gastric and intestinal conditions. Our findings suggest that CSNPs can be used as an effective nanocarrier to encapsulate anthocyanins exacted from red-fleshed apples, and XJ4-CSNPs have the advantage in anthocyanins applications compared to RL-CSNPs.

## Figures and Tables

**Figure 1 molecules-24-03366-f001:**
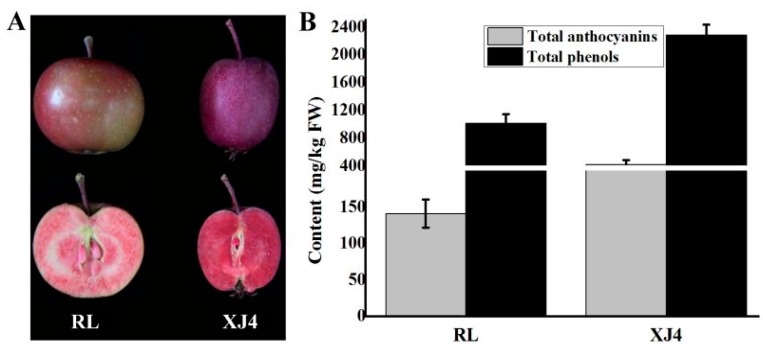
The phenotype, total anthocyanins, and phenols content of two different types of red-fleshed apples. (**A**) Intact and longitudinal section of Red Laiyang (RL) and Xinjing No.4 (XJ4); (**B**) total anthocyanins and phenols content of RL and XJ4.

**Figure 2 molecules-24-03366-f002:**
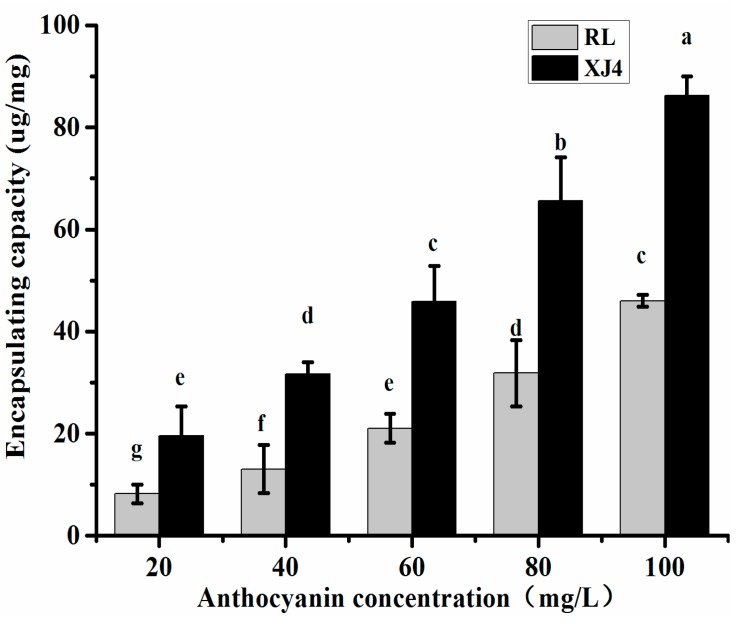
Encapsulating capacity of Corn Starch Nanoparticles (CSNPs) under different anthocyanin concentrations. Within samples, different letters (a–g) mean significant differences (*p* < 0.05).

**Figure 3 molecules-24-03366-f003:**
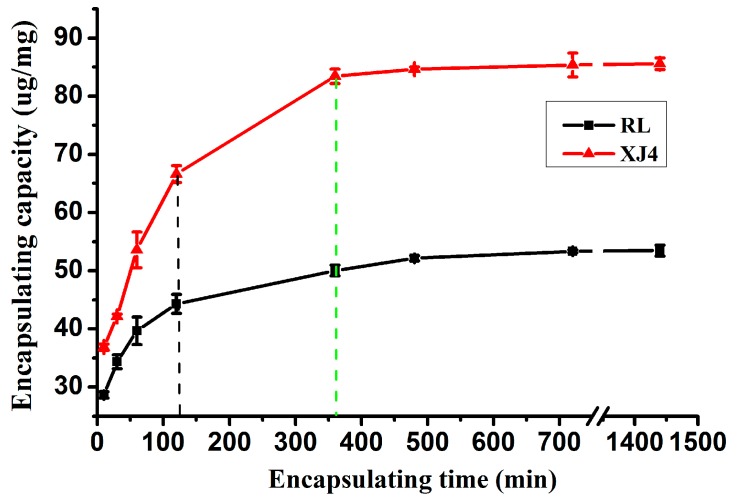
Encapsulating capacity of CSNPs under various encapsulating times of anthocyanins.

**Figure 4 molecules-24-03366-f004:**
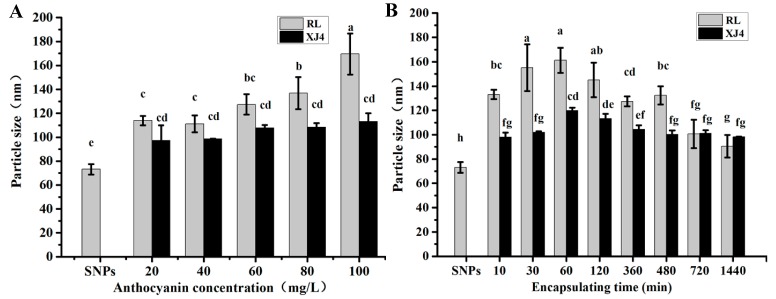
Size of CSNPs under various anthocyanin concentrations and encapsulating times. Within samples, different letters (a–e for (**A**), a–h for (**B**)) were significantly different (*p* < 0.05).

**Figure 5 molecules-24-03366-f005:**
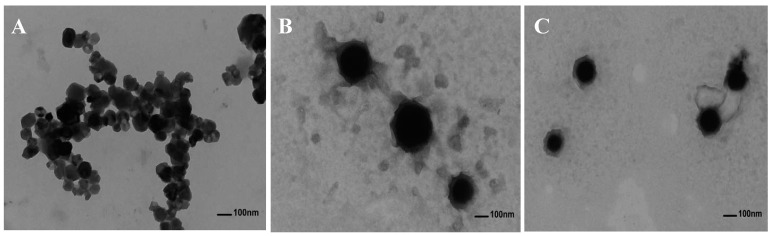
Transmission electron microscopy images of nanoparticles. (**A**) CSNPs; (**B**) RL-CSNPs; (**C**) XJ4-CSNPs.

**Figure 6 molecules-24-03366-f006:**
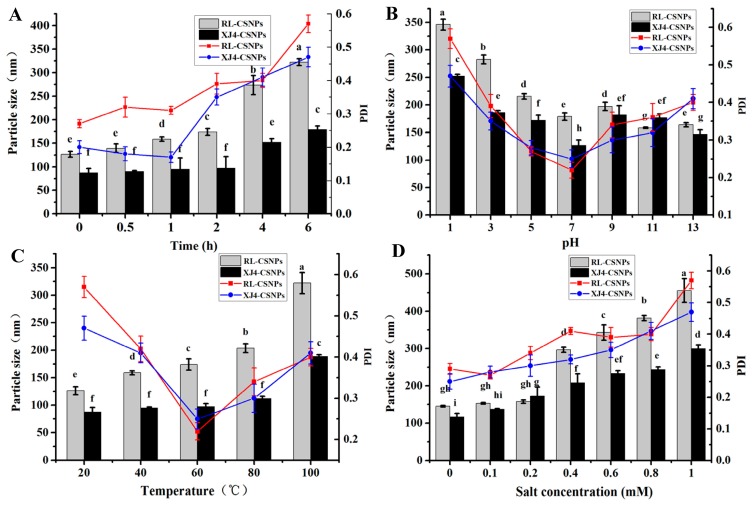
The size and PDI of anthocyanin nanoparticles in different stress conditions. (**A**) Ultraviolet light exposure time; (**B**) pH; (**C**) temperature; (**D**) salt. The bar chart represents the distribution of particle size and the line chart represents the distribution of PDI. Within samples, different letters (a–f for Figure 6A, a–h for Figure 6B, a–f for Figure 6C, a–i for Figure 6D) were significantly different (*p* < 0.05).

**Figure 7 molecules-24-03366-f007:**
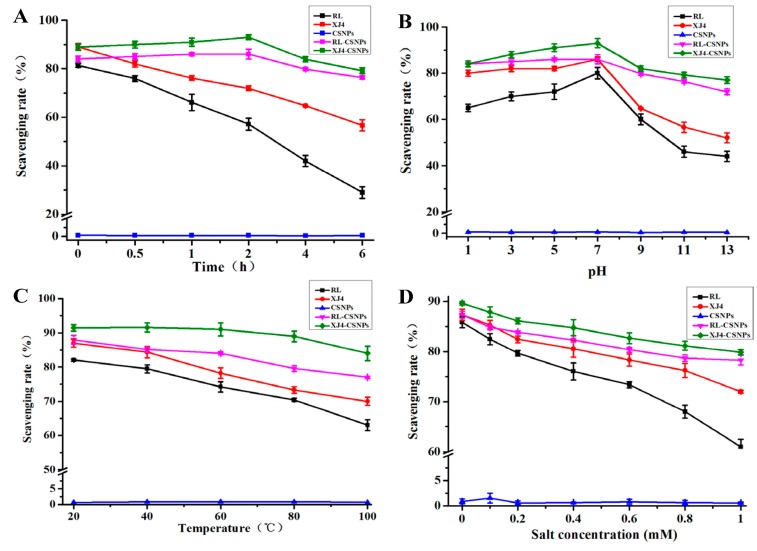
DPPH free radical scavenging ability of nanoparticles in different stress conditions. (**A**) Ultraviolet light exposure time; (**B**) pH; (**C**) temperature; (**D**) salt.

**Figure 8 molecules-24-03366-f008:**
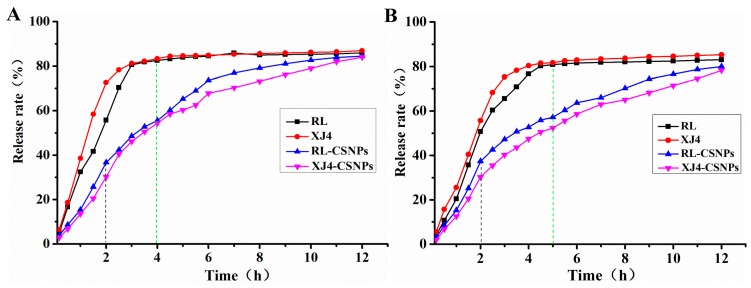
In vitro release rates of anthocyanins before and after nanocrystallization. (**A**) Simulated gastric (pH 2.0) condition; (**B**) simulated intestinal (pH 7.0) condition.

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
