# Peer review of "Effect of Nanocrystallization of Anthocyanins Extracted from Two Types of Red-Fleshed Apple Varieties on Its Stability and Antioxidant Activity"

_molecules, 2019, doi:10.3390/molecules24183366_

Round 1

Reviewer 1 Report

The authors conducted a careful study of anthocyanins extracted from two different types of red-fleshed apples after encapsulation onto corn starch nanoparticles. The study concerned both the morphology of the nanoparticles and their behaviour when subjected to different types of stress (e.g. UV radiation, temperature, salinity). The paper addresses a topic of great interest, is well-structured and well-organized, and should be of interest and assistance to scholars in the field. However, the manuscript contains many formal and typographical errors which appear to be caused by word-processing software bugs. Thus, the authors need to proofread the manuscript more thoroughly in order to make it suitable for publication.

line 40:  Malus sieversii f. neidzwetzkyana

line 47: ...and cancer...

line 49: ....with proteins to...

line 52: ....or storage....

line 56: ...in our previous study....

line 70: ...significantly enhanced stability....

line 78: ...of red-fleshed....

line 83: ...regard to using....

Reviewer 2 Report

Authors described that the differences in CNSPs encapsulating for anthocyanins between RL and XJ4 might be caused by differences in the metabolites (Page 4 line 137 -139). Please explain the role of the metabolites and which compounds in the metabolites. Given these finding, we hypothesis that XJ4-CSNPs could have stronger antioxidant activity (Page 6 line 218 -219). Why? There should be many reasons for this result, such as particle size, molecular interactions …. The unique compounds observed in XJ4 such as cyanidin and syringetin glycosides might explain why XJ4-CSNPs antioxidant activity was stronger. Why? Please explain that the antioxidant activities and interaction with CNSPs of cyanidin and syringetin glycosides. Please provide the meanings of statistical symbols in the figures. Please provide the repeated number of the data in the figures. Please provide the standard deviations in the figure 3. The text described in the manuscript is inconsistent with the figure legend of figure 1. Please provide the methods of assay metabolite compounds in the supplementary data.

Reviewer 3 Report

The Authors propose new results with great potential.

The quality of the presentation is high.

The results are clearly exposed and the conclusions quite logical.

The Materials and Methods can be improved, in particular by providing detailed description of the analytical method used for Supplemental Table.

The Figures are not readable in print (too small) in their present forms.

Round 2

Reviewer 2 Report

The author has responded and revised all comments. This manuscript is acceptable for publication in this journal "Molecules".